# Deep Insights into Noisy Pseudo Labeling on Graph Data

**Botao Wang**[1,2], **Jia Li**[1,2*], **Yang Liu**[1,2], **Jiashun Cheng**[1,2],
**Yu Rong**[3], **Wenjia Wang**[1,2], **Fugee Tsung**[1,2]
[1]Hong Kong University of Science and Technology, Hong Kong SAR, China
[2]Hong Kong University of Science and Technology (Guangzhou), Guangzhou, China
[3]Tencent AI Lab, Shenzhen, China
{bwangbk, yliukj, jchengak}@connect.ust.hk
{jialee, wenjiawang season}@ust.hk, yu.rong@hotmail.com

## Abstract

Pseudo labeling (PL) is a wide-applied strategy to enlarge the labeled dataset by self-annotating the potential samples during the training process. Several works have shown that it can improve the graph learning model performance in general. However, we notice that the incorrect labels can be fatal to the graph training process. Inappropriate PL may result in the performance degrading, especially on graph data where the noise can propagate. Surprisingly, the corresponding error is seldom theoretically analyzed in the literature. In this paper, we aim to give deep insights of PL on graph learning models. We first present the error analysis of PL strategy by showing that the error is bounded by the confidence of PL threshold and consistency of multi-view prediction. Then, we theoretically illustrate the effect of PL on convergence property. Based on the analysis, we propose a cautious pseudo labeling methodology in which we pseudo label the samples with highest confidence and multi-view consistency. Finally, extensive experiments demonstrate that the proposed strategy improves graph learning process and outperforms other PL strategies on link prediction and node classification tasks.

## 1 Introduction

Pseudo Labeling (PL) [27, 13] is one of the most popular self-supervised learning approaches and has been widely used to tackle the label sparsity problem. Its core idea is to enlarge the training set by self-labeling. As most self-labeled samples should be consistent with ground truth, the enlarged dataset has a larger sample capacity to improve the model generalization. Plenty of studies have shown the effectiveness of PL [7, 26, 14, 5]. However, there is a trade-off between the benefit of PL and the effect of mislabeled samples. When the benefit of PL outweighs the impact of introduced noise, the performance of the base model (i.e., without PL) can be improved. But for non-i.i.d. condition such as graph data, the introduced noisy labels may transfer among the samples and be amplified, which may degrade the performance of base model. Although several methods have been proposed to alleviate this noisy phenomenon [31, 32, 15], there is still neither a clear explanation nor a quantification of how pseudo labeling affects the graph learning models.

In this study, we attempt to answer questions above. Specifically, we evaluate PL's effect on the prediction error of the base model and the convergence of the empirical loss function. For a graph learning model, the message aggregation process would amplify the noises of incorrect labels introduced by PL. These noises can even accumulate to damage the base model's performance. For

---

*corresponding author

37th Conference on Neural Information Processing Systems (NeurIPS 2023).

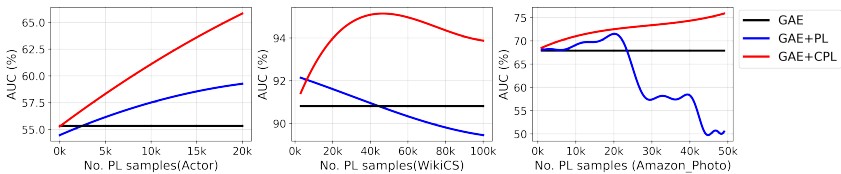

Figure 1: The performance of GAE, PL and proposed CPL strategy on link prediction w.r.t. Actor, WikiCS and Amazon_Photo dataset. Horizontal axis is #PL samples, and vertical axis is AUC.

example, in a two-layer graph neural network (GNN) for node classification, the mislabeled nodes can inappropriately reduce the predicted confidence of their 2-hop neighbors. Moreover, in some circumstances, such as link prediction task, the pseudo labeled sample would not only serve as the label but also as the model inputs in the consequent iterations. This characteristic further aggravates the loss function's convergence due to the noises added to adjacency matrix.

To visualize the side effect of the PL on graph data, we conduct a toy experiment as shown in Figure 1, where the benefit of PL to a popular link prediction model **GAE** [12] depends on the choice of graph dataset, i.e., PL improves the performance of GAE on Actor, but degrades that on WikiCS. Even worse, due to the incorrect label introduced by PL, PL leads to the model's collapse on Amazon_Photo.

To obtain a stable and consistent result by PL strategy, it is necessary to quantify the impact of introduced noisy labels and theoretically analyze how it affects the graph training procedure compared to the procedure without PL. In this paper, we build a theoretical connection between PL strategy and graph learning models with multi-view augmentations. We prove that the error bound of the PL predictor is jointly bounded by a confidence threshold and the prediction consistency over different augmentations. Moreover, we theoretically analyze that PL strategy affects convergence property by the covariance term in the optimization function. Accordingly, we propose the Cautious Pseudo Labeling (CPL) strategy for the graph learning process that maximizes the confidence threshold by committing the PL samples with high prediction probability. We evaluate the CPL on different graph learning tasks, including link prediction and node classification models, where we observe remarkable and consistent improvements over multiple datasets and base models. As shown in Figure 1, compared with the base model GAE, the average AUC improvement of CPL over three datasets is 7.79%, which clearly validates the superiority of our model.

## 2   Model

In this section, we define the general problem of graph learning, including link prediction and node classification tasks. We give the error bound and convergence property analysis associated with the application of PL strategy. Lastly, we propose cautious PL strategy accordingly.

### 2.1   Problem Definition

In graph learning, given graph $G = (\mathcal{V}, \mathcal{E})$, $\mathcal{V} = \{v_i\}$ is the node set, $\mathcal{E} = \{(i, j)\}$ is the edge set, and $|\mathcal{V}| = N$ is the node number. The feature matrix and adjacent matrix are denoted by $X = [x_{ij}]_{N \times F}$ and $A = [a_{ij}]_{N \times N}$, respectively, where $(i, j) \in \mathcal{E}$ if and only if $a_{ij} \neq 0$. A base GNN model $g$ is trained on the graph $G$ and the observed labels. It outputs the probability of prediction target. In this work, we adopt the PL scheme which involves a teacher model $g_\phi$ and a student model $g_\psi$. The teacher model calculates confidence of the unlabeled samples and PL a subset of samples using the strategy $\mathcal{T}$. Subsequently, student model utilizes the enlarged set to fine-tune the base model and becomes the teacher model in the next iteration.

**Node classification task.** The objective of the node classification task is to predict the probabilities of an unlabeled node belonging to different classes $g : G \to \mathbb{R}^{N \times M}$, where $M$ is the number of classes. The original training set comprises the labeled nodes $\mathcal{V}_o$ and their labels $Y_o$. And the model aims to predict the classes of the unlabeled nodes $\mathcal{V}_u$. In the PL scheme, the teacher model calculates the confidence of the unlabeled nodes $\hat{Y}_u$ and assigns pseudo labels to a selected subset of nodes

$\{\mathcal{V}_p, Y_p | \mathcal{V}_p \subset \mathcal{V}_u, Y_p \subset \hat{Y}_u\}$. Then the student model undergoes fine-tuning on the enlarged label set $g_\psi(G) \to \hat{Y}_o$, $\hat{Y}_o = Y_o \cup Y_p$, $\hat{Y}_o$ is the enlarged label set.

**Link prediction task.** The prediction targets are the probabilities of edges between unlabeled node pairs. In our scheme, the link prediction model $g(g_{emb}, s)$ consists of an embedding network $g_{emb} : G \to \mathbb{R}^{N \times D}$ and a score function $s : \mathbb{R}^D \times \mathbb{R}^D \to \mathbb{R}$ that estimates the probability of the link, where $D$ is the dimension of the node embeddings. A portion of edges is observed as the training graph $G = (\mathcal{V}, \mathcal{E}_o), \mathcal{E}_o \subset \mathcal{E}$. The teacher model predicts the confidence of the unobserved edges $\mathcal{E}_u$ and enlarge the observed edge set $\hat{\mathcal{E}}_o = \mathcal{E}_p \cup \mathcal{E}_o$ for fine-tuning $g_\psi : G(\mathcal{V}, \hat{\mathcal{E}}_o) \to \mathbb{R}^{N \times N}$.

It is worth to mention that in the link prediction task, the node embeddings undergo a change once pseudo labels are added, even before fine-tuning. Because the enlarged edge set is also utilized as input for the GNN.

The optimization target is formulated as minimizing $\mathcal{L} = \text{CE}(g_\psi, Y_o)/|Y_o|$, where $\text{CE}(\cdot)$ denotes cross entropy, $Y_o$ is the ground truth label of the observed set and represents $\mathcal{E}_o$ for the link prediction task. The overall performance is evaluated by the 0-1 loss: $\text{Err}(g) = \mathbb{E}[\text{argmax}(g_\psi) \neq Y]$.

## 2.2 Pseudo Labeling Error Analysis

We here present the error bound analysis of the graph learning under the PL strategy. To facilitate mathematical treatment, we introduce several assumptions.

### 2.2.1 Graph perturbation invariant

We assume **graph perturbation invariant** (GPI) property in GNN, which states that the prediction variance is linearly bounded by the difference between the augmented and original inputs.

**Definition 2.1**: *Given a graph $G$ and its perturbation $\hat{G} = G(X \odot M_x, A \odot M_a)$ by the random feature masks $M_x \in \{1, 0\}^{N \times F}$ and adjacent matrix mask $M_a \in \{1, 0\}^{N \times N}$ satisfying*

$$\frac{1}{N \cdot F}\|\mathbf{1}^{N \times F} - M_x\|_2^2 + \frac{1}{N^2}\|\mathbf{1}^{N \times N} - M_a\|_2^2 < \epsilon, \tag{1}$$

*the GNN $g(\cdot)$ has GPI property if there exists a constant $C > 0$ such that the perturbed prediction confidence satisfies $\|g(\hat{G}) - g(G)\|_2^2 < C\epsilon$. Here, $\odot$ is element-wise product, and $\|\cdot\|_2$ is the 2-norm of the vector or matrix.*

The GPI property guarantees the variation of the output confidence is linearly bounded by the degree of graph perturbation, which is similar to the $C$-Lipschitz condition applied to GNN.

### 2.2.2 Additive expansion property

With Definition 2.1, if we have a convex measurable function $f(y)$ that satisfies the $C$-Lipchitz condition, we can find a probability measure on the output prediction $p_f(y)$ that satisfies the **additive expansion property** as stated in Proposition 2.2.

**Proposition 2.2** *Define a local optimal subset as $U \subset Y$, whose probability is higher than a threshold $p_f(y) > 1 - q, y \in U$, and its perturbation set $U_\epsilon = \{\hat{y} = g(G) : \|\hat{y} - y\|_2 \leq C\epsilon, y \in U\}$, where $G \in \{\hat{G}\}$ is the space of the perturbed graph. Then, there exists $\alpha > 1, \eta > 0$, s.t. the probability measure $p_f$ satisfying following additive expansion property: $p_{\alpha f}(U_{\hat{\epsilon}} \setminus U) \geq p_{\alpha f}(U) + \eta \cdot \alpha$.*

The proposition guarantees the continuity of the measurable function in the neighborhood of the local optimal subset $U$. In practice, $U$ represents the PL samples, which ideally should be close to the ground truth labels in ideal. However, the correctness of the PL samples is discrete where the continuity condition is hard to be satisfied. To address this issue, we can leverage GPI. By applying multi-view augmentations and calculating average confidence, we canreparameterize the probability measure to be continuous. In this condition, the Proposition 2.2 implies that the probability of the neighborhood under the amplified measure is greater than the original local optimum. This observation opens up potential opportunities for optimization. For a detailed proof of Proposition 2.2, please refer to Appendix A.

### 2.2.3 Prediction error measurement

Then, we use the additive expansion property to calculate the error bound of the multi-view GNN model under PL strategy. According to Theorem B.2 in [25], when we use 0-1 loss to measure the error of predictor, the error bound can be evaluated by the following theorem.

**Theorem 2.3** *Let $q > 0$ be a given threshold. For the GNN in the teacher model $g_\phi$, if its corresponding density measure satisfies additive expansion, the error of the student predictor $g_\psi$ is bounded by*

$$\text{Err}(g) \leq 2(q + \mathcal{A}(g_\psi)), \tag{2}$$

*where $\mathcal{A}(g_\psi) = \mathbb{E}_{Y_{test}}[\boldsymbol{I}(\exists g_\psi(\hat{G}) \neq g_\psi(G))]$ measures the inconsistency over differently augmented inputs, $Y_{test}$ is the test set for evaluation.*

The brief derivation is provided in Appendix.B. From to Eq.2, we observe that the prediction error (0-1 loss) is bounded by the confidence threshold $q$ and the expectation of the prediction inconsistency $\mathcal{A}$. Intuitively, if the predictions of multi-view augmented inputs tend to be consistent and the threshold is smaller, the error bound will be smaller, which implies a more accurate predictor.

If $q$ is a small value, the threshold probability of PL $1 - q$ approaches 1. This cautious approach in self-training leads to a smaller lower bound in the error estimate. When applying random PL, setting the confidence threshold as $q = 0.5$, then the maximum theoretical error rate is 1. It means that there is no guarantee on the performance after applying PL.

The inconsistency term $\mathcal{A}$ is the expectation of the probability that predictions are inconsistent when input graphs are augmented differently. A small value of $\mathcal{A}$ indicates consistent prediction across different views. In such cases, we have more confidence in the predictions, leading to a smaller error bound. On the other hand, if the predictions from different views are inconsistent with each other, the model lacks robustness, resulting in a larger error bound.

## 2.3 Convergence Analysis

In this section, we analyze the influence of the PL strategy on the empirical loss to illustrate its convergence property. First, we assume that optimizing teacher model will not influence the convergence property of PL strategy. This assumption stems from the understanding that the teacher model can converge independently without the incorporation of PL.

**Assumption 2.4** *The loss function defined in Algorithm 1 is non-increasing during the optimization:* $CE(g_\psi^{(t)}, \hat{Y}_o^{(t+1)}) \leq CE(g_\phi^{(t)}, \hat{Y}_o^{(t)})$.

Then, we show that the PL sample selection strategy $\mathcal{T}$ influences the covariance term derived from the empirical loss, then affects the convergence property.

$$\mathcal{L}_{\mathcal{T}}^{(t+1)} \leq \beta \text{Cov}\left[\text{ce}\left(g_\psi, Y\right), \mathcal{T}\right] + \mathcal{L}_{\mathcal{T}}^{(t)} \tag{3}$$

where $\beta = |\hat{Y}_u|/(|\hat{Y}_o| + k)$, $ce(\cdot)$ is the element-wise cross entropy. The equality is achieved when the model reaches the global optimal under given training set.

The detailed derivation of the Eq.3 is shown in Appendix.C. From Eq.3, we observe that the effect of PL strategy is decoupled and encapsulated in the covariance term. The covariance sign determines whether the empirical loss will increase or decrease in the next iteration of optimization. For instance, when PL samples are randomly selected, $\mathcal{T}$ would be independent with $g_\psi$. The covariance becomes 0, indicating that the PL does not influence the loss function. However, a carefully designed PL strategy can accelerate the optimization of empirical loss and yield improved convergence properties.

## 2.4 Cautious Pseudo Labeling

According to Theorem 2.3, setting a higher confidence threshold for pseudo labeling (PL) can lead to a lower prediction error. This is reflected in a positive covariance term, which accelerates the optimization of the empirical loss and improves the convergence property. In order to satisfy these requirements, we propose the iterative Cautious Pseudo Labeling (CPL) strategy. CPL involves carefully and iteratively pseudo labeling the most confident samples to maximize these improvements in prediction accuracy and convergence.

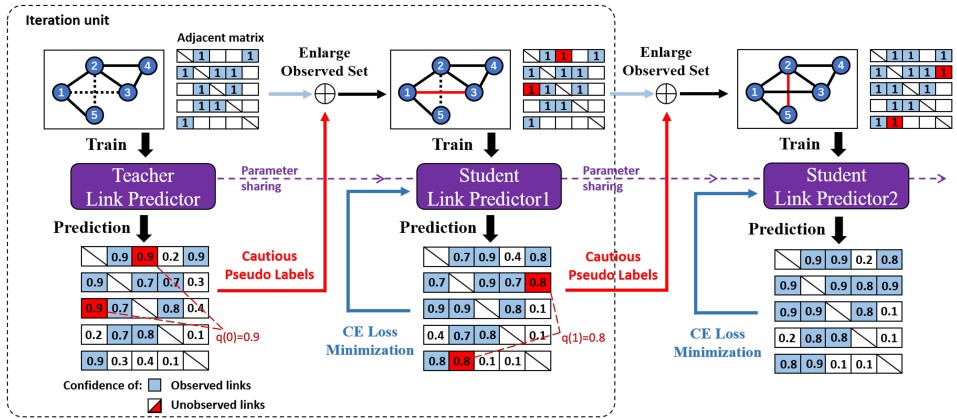

Figure 2: A framework illustration of CPL on link prediction: There is a teacher model and a student model that share parameter. The most confident samples in teacher model's prediction are pseudo labeled. Then the student model is fine-tuned on the enlarged dataset and becomes teacher model in the next iteration.

There are a teacher model and a student model in each iteration. First, we calculate multi-view prediction by the teacher model $g_\phi$ as the confidence measure. Although biased, it is a simple and effective measure in many PL strategies. In node classification, the GNN directly outputs the probability as confidence $g(G)$. For link prediction, the confidence is determined by the inner product similarity of node embedding $g(G) = \sigma(E^T E)$, where $E = (e_1, ..., e_N)^T = g_{emb}(G)$ is the node embedding, and $\sigma(\cdot)$ is the sigmoid function.

Then we select PL samples in unobserved set with the highest confidence, enlarging the training set. We iteratively select top-$k$ confident samples for PL: $Y_p^{(t)} = \mathcal{T}(\hat{Y}_u^t, \bar{g}_\phi, k)$, where $\mathcal{T} \in \{0,1\}^{|\hat{Y}_u^{(t)}|}, \sum \mathcal{T} = k$. $\bar{g}_\phi^{(t)}$ is the averaged confidence of the multi-view augmentation and is equal to $g_\phi^{(t)}$ for the single input. At the $t$-th iteration, the selected PL samples are $Y_p^{(t)}$.

Then we update the observed and the unobserved set. The student model is fine-tuned by the enlarged set and becomes the new teacher model in the next iteration. We take the link prediction task as the example, whose main scheme is in Fig.2. The complete algorithm of CPL is shown in Algorithm 1.

The confidence threshold $q(t)$ recorded in Algorithm 1 is the lowest confidence among the PL samples $Y_p^{(t)}$. At each iteration, we record the lowest confidence in these $k$ PL samples as $q^{(t)}$. We update $q$ to be $q^{(t)}$ if $q^{(t)}$ is smaller. Finally, $q$ serves as the confidence threshold in Eq.2 for error analysis in Theorem 2.3.

The following theorem states the improvement on convergence property under CPL.

**Theorem 2.5** *Let $\mathcal{L}_\mathcal{T}^{(t)}$ denote the optimization target at the $t$-th iteration. The risk is monotonically decreasing under pseudo labeling strategy, i.e.,*

$$\mathcal{L}_\mathcal{T}^{(t+1)} \leq \beta \text{Cov}\left[\text{ce}\left(g_\psi, Y\right), \mathcal{T}\right] + \mathcal{L}_\mathcal{T}^{(t)} \leq \mathcal{L}_\mathcal{T}^{(t)}. \tag{4}$$

*Here $\mathcal{L}_\mathcal{T}^{(t+1)}$ is calculated by $1/|\hat{Y}_o^{(t+1)}|CE(g_\psi^{(t)}, \hat{Y}_o^{(t+1)})$.*

We have demonstrated the first inequality in Eq.3. The second inequality holds because the covariance between the cross-entropy loss and PL strategy is non-positive. On one hand, each view of $g_\psi$ is positively correlated to the ground-truth label $Y$ due to the consistency regularization. Then, $g_\psi$ exhibits negative correlation with the cross-entropy loss $\text{ce}(g_\psi, Y)$. On the other hand, the PL strategy $\mathcal{T}$ can be seen as a discrete sign function. In CPL, higher confidence samples are more likely to be pseudo-labeled. Therefore, $\mathcal{T}$ is positively correlated with $g_\psi$. Consequently, $\mathcal{T}$ is negatively correlated with the cross-entropy term, resulting in a non-positive covariance term $\text{Cov}(\cdot)$. This guarantees the monotone decreasing behavior of the loss function, as stated in Theorem 2.5. Furthermore, since the loss function is lower bounded, Theorem 2.5 ensures the convergence of Algorithm 1.

**Algorithm 1:** Iterative cautious pseudo labeling.

---

**Input:** Graph $G$, observed and unobserved label set $Y_o, Y_u$, iterative and total pseudo labeling number $k, K$

**Output:** Student model $g_\psi(G)$, confidence threshold $q$

Pre-train teacher model $g_\phi^{(0)}$ on observed set $\{G, Y_o\}$ ;

Initialize student model: $g_\psi^{(0)} = g_\phi^{(0)}$;

**while** $\left|\hat{Y}_o^{(t)}\right| \leq K$ **do**

    Calculate the average confidence of the unobserved set by the multi-view teacher model $g_\phi$.;

    Select pseudo labeling subset $Y_p^{(t)}$ with top-$k$ confidence in $\hat{Y}_u^{(t)}$;

    Update set: $\hat{Y}_o^{(t+1)} = \hat{Y}_o^{(t)} \cup Y_p^{(t)}, \hat{Y}_u^{(t+1)} = \hat{Y}_u^{(t)} \setminus Y_p^{(t)}$ ;

    Update the current confidence threshold $q$ according to $q(t)$;

    Fine-tune the student model $g_\psi^{(t)}$ by minimizing the cross entropy on $\hat{Y}_o^{(t+1)}$;

    Update teacher model $g_\phi^{(t+1)} = g_\psi^{(t)}$ and set $g_\psi^{(t+1)} = g_\phi^{(t+1)}$;

    $t = t + 1$;

**end**

---

**Time complexity** The computational complexity of CPL depends on the complexity of specific GNN models used. The key operation in CPL is the selection of top-$k$ values. For node classification, it takes $O(N\log k)$, while for link prediction, it takes $O(N^2\log k)$. However, since GNN models typically compute probabilities for all samples, the overhead introduced by CPL does not increase the complexity of existing methods significantly. For example, the complexity of GAE [12] for link prediction is $O(|\mathcal{E}_o|D^2 + N^2 D)$ (i.e., the complexity of graph convolution operations and inner-products). Integrating CPL into GAE does not increase the complexity since both involve $O(N^2)$ operations. In practice, the additional time consumption mainly results from the fine-tuning process.

## 3 Experiments

In this section, we conduct an evaluation to assess the effectiveness of the CPL strategy on both link prediction and node classification tasks. We compare CPL with raw base models as well as other PL strategies. Then, we analyze the impact of CPL capacity, training data ratio, PL strategy, and augmentation methods. Finally, a case study is bridged to the theoretical analysis on convergence and error bound. The implementation is open-sourced at https://github.com/AcEbt/CPL.

### 3.1 Datasets and Benchmarks

We adopt five public available datasets to evaluate CPL strategy for link prediction, i.e. CiteSeer, Actor, WikiCS, TwitchPT, and Amazon_Photo, and five datasets for node classification, i.e. Cora, CiteSeer, PubMed, Amazon_Photo, and LastFMAsia. Detailed statistics are reported in Table 1.

In link prediction task, as there are few PL-based methods, we apply the CPL strategy on three popular models: **GAE** [12],**node2vec** [4], **SEAL** [29] . To reserve sufficient candidate unobserved samples for PL, the dataset is randomly split into 10%,40%,50% for training, validation, and testing.

In node classification task, we employ CPL on 4 popular base models: **GCN**, **GraphSAGE**, **GAT**, **APPNP**. CPL is compared with two other PL strategies, namely **M3S**[22] and **DR-GST**[18], using the implementation and parameters provided by the authors [2]. We adopt the official split for the citation datasets and 5%,15%,80% split for other datasets in node classification.

We run experiments with 5 random seeds and report the mean and standard deviations of the metrics.

---

[2]https://github.com/BUPT-GAMMA/DR-GST

Table 1: Dataset statistics.

| Dataset | Cora | CiteSeer | PubMed | Actor | WikiCS | TwitchPT | Amazon_Photo | LastFMAisa |
|---|---|---|---|---|---|---|---|---|
| # Nodes | 2,078 | 3,327 | 19,717 | 7,600 | 11,701 | 1,912 | 7,650 | 7,624 |
| # Links | 10,556 | 9,104 | 88,648 | 30,019 | 216,123 | 64,510 | 238,162 | 55,612 |
| # Features | 1433 | 3,703 | 500 | 932 | 300 | 128 | 745 | 128 |
| # Classes | 7 | 6 | 3 | 5 | 10 | 2 | 8 | 18 |

Table 2: Performance (AUV%) comparison on link prediction.

| | Model | Citeseer | Actor | WikiCS | TwitchPT | Amazon_Photo |
|---|---|---|---|---|---|---|
| **AUC(%)** | GAE | 71.10 ± 0.56 | 55.34 ± 0.57 | 90.81 ± 0.69 | 74.48 ± 3.03 | 67.92 ± 1.31 |
| | GAE+CPL | **72.45 ± 0.24** | **65.58 ± 1.04** | **95.56 ± 0.24** | **79.67 ± 3.77** | **76.30 ± 1.84** |
| | node2vec | 52.03 ± 0.60 | 53.30 ± 0.59 | 88.82 ± 0.28 | 79.46 ± 0.77 | 89.32 ± 0.21 |
| | node2vec+CPL | **55.22 ± 1.63** | **65.11 ± 2.31** | **91.99 ± 0.26** | **84.76 ± 3.52** | **89.53 ± 0.30** |
| | SEAL | 63.60 ± 0.01 | 73.41 ± 0.02 | 86.01 ± 0.04 | 87.80 ± 0.01 | 76.96 ± 0.17 |
| | SEAL+CPL | **64.33 ± 0.14** | **73.54 ± 0.01** | **86.83 ± 0.07** | **87.87 ± 0.01** | **78.86 ± 0.01** |
| **AP(%)** | GAE | 72.12 ± 0.63 | 53.60 ± 1.06 | 90.58 ± 0.71 | 69.73 ± 5.06 | 67.06 ± 0.99 |
| | GAE+CPL | **73.54 ± 0.20** | **67.65 ± 1.06** | **95.58 ± 0.29** | **79.09 ± 5.48** | **75.52 ± 4.23** |
| | node2vec | 52.90 ± 0.36 | 55.43 ± 0.62 | 92.54 ± 0.51 | 83.37 ± 0.52 | 91.46 ± 0.18 |
| | node2vec+CPL | **56.19 ± 1.60** | **68.33 ± 2.85** | **93.66 ± 0.29** | **85.87 ± 2.15** | **91.47 ± 0.21** |
| | SEAL | 64.38 ± 0.01 | 73.17 ± 0.12 | 83.63 ± 0.16 | 87.69 ± 0.01 | 73.72 ± 0.56 |
| | SEAL+CPL | **64.94 ± 0.14** | **73.44 ± 0.02** | **86.72 ± 0.12** | **87.75 ± 0.02** | **80.36 ± 0.09** |

## 3.2 Performance Comparison

### 3.2.1 Overall performance comparison

For link prediction, Table 2 lists the AUC and AP of raw baselines and ones with CPL employed. We observe that CPL distinctively increases the performance of baseline models in nearly all cases. Note that the CPL strategy can achieve performance gain under the circumstances of both high and low performance (e.g., the AUC of GAE on Actor improves from 55.34 to 65.58 and the AP of SEAL on CiteSeer improves from 64.38 to 64.94).

For node classification, Table 3 shows the improvement on the base models and comparison with other PL strategies. The CPL strategy can consistently improve the performance of the base models. However, other PL strategy may be ineffective or degrade the base model (e.g., DR-GST on Cora with APPNP, M3S for PubMed with GraphSAGE). CPL also consistently outperforms other PL strategies.

The time consumption is also reported in AppendixD.

### 3.2.2 Impact of pseudo labeling capacity

In the experiment, the number of PL samples in each iteration $k$ is set from 100 to 1000. We provide a reference interval for the selection of $k$. Intuitively, a small $k$ can lead to a reliable result. But a too small $k$ will unnecessarily increase the training time. And it does not prominently influence the overall performance as we test on different datasets. In the experiment, the predicted confidence distribution is around 1, as there are usually plenty of potential PL samples. $k$ is much smaller than the total number of unobserved samples. Then the $k$-th highest confidence in the unobserved set will not have much difference, as the cases in Table 4.

### 3.2.3 Impact of training data ratio

PL enlarges the observed dataset and introduces extra information for the training. This effect has a different degree of contribution on different training data. When the training set is partially applied for the training, the ratio of the observed set ranges from 0.1 to 0.9. The variation of AUC and AP on the Amazon_Photo dataset is shown in Fig.3. The CPL method can consistently improve the performance of the raw model even starting from a small training set. It is also worth to mention that CPL is more likely to have a more significant contribution when the training set is small, as there is more introduced information.

Table 3: Performance (AUV%) comparison on node classification.

| Model | | Cora | CiteSeer | PubMed | Amazon_Photo | LastFMAsia |
|---|---|---|---|---|---|---|
| **GCN** | Raw | 80.74 ± 0.27 | 69.32 ± 0.44 | 77.72 ± 0.46 | 92.62 ± 0.45 | 78.53 ± 0.60 |
| | M3S | 80.92 ± 0.74 | 72.70 ± 0.43 | 79.36 ± 0.64 | 93.07 ± 0.25 | 79.49 ± 1.42 |
| | DR-GST | 83.54 ± 0.81 | 72.04 ± 0.53 | 77.96 ± 0.25 | 92.89 ± 0.16 | 79.31 ± 0.55 |
| | CPL | **83.94 ± 0.42** | **72.96 ± 0.22** | **79.98 ± 0.92** | **93.15 ± 0.24** | **79.92 ± 0.61** |
| **GraphSAGE** | Raw | 81.12 ± 0.32 | 69.80 ± 0.19 | 77.52 ± 0.38 | 92.46 ± 0.17 | 80.23 ± 0.28 |
| | M3S | 83.02 ± 0.49 | 70.98 ± 2.14 | 79.12 ± 0.25 | 92.41 ± 0.14 | 81.48 ± 0.56 |
| | DR-GST | 81.02 ± 1.99 | 72.28 ± 0.35 | 76.96 ± 0.43 | 92.58 ± 0.14 | 81.10 ± 0.30 |
| | CPL | **84.62 ± 0.19** | **73.14 ± 0.21** | **79.72 ± 0.72** | **92.90 ± 0.20** | **82.25 ± 0.25** |
| **GAT** | Raw | 81.28 ± 0.87 | 71.18 ± 0.43 | 77.34 ± 0.34 | 93.26 ± 0.31 | 81.12 ± 0.58 |
| | M3S | 82.28 ± 0.95 | 71.70 ± 0.72 | 79.20 ± 0.21 | 93.71 ± 0.16 | 81.82 ± 0.93 |
| | DR-GST | 83.32 ± 0.31 | 72.64 ± 0.97 | 78.28 ± 0.32 | 93.60 ± 0.13 | 81.86 ± 0.50 |
| | CPL | **83.86 ± 0.22** | **73.02 ± 0.37** | **79.62 ± 0.31** | **93.72 ± 0.29** | **82.89 ± 0.56** |
| **APPNP** | Raw | 82.52 ± 0.69 | 70.82 ± 0.24 | 79.96 ± 0.50 | 93.05 ± 0.29 | 82.40 ± 0.50 |
| | M3S | 82.54 ± 0.40 | 72.58 ± 0.45 | 79.98 ± 0.14 | 93.21 ± 0.59 | 83.55 ± 0.71 |
| | DR-GST | 82.46 ± 0.87 | 72.64 ± 0.54 | 80.00 ± 0.48 | 93.12 ± 0.32 | 82.88 ± 0.35 |
| | CPL | **84.20 ± 0.42** | **74.22 ± 0.24** | **80.62 ± 0.24** | **93.48 ± 0.23** | **83.56 ± 0.53** |

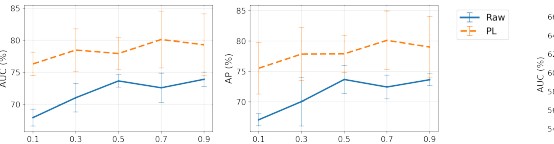
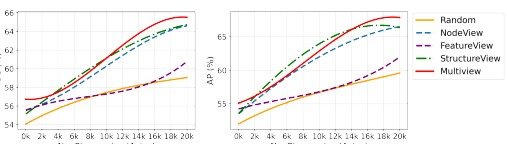

Figure 3: The effect of training ratio on Amazon_Photo w.r.t. AUC and AP: the relative improvement is more significant when the training ratio is small (i.e., the observed graph is sparse).

Figure 4: The effect of different data augmentation strategy on Actor w.r.t. AUC and AP: the improvement of multi-view is largest compared to other methods.

### 3.2.4 Impact of noisy pseudo labeling strategy

As the noise introduced by PL has a large effect in the graph, we compare the CPL with the PL strategy. In PL, the samples are selected randomly in the unobserved set, whose labels are estimated by the teacher model. The comparison of AUC's variation on different datasets is shown in Fig.1. We discover that PL has different effects on the baseline models. On Actor, the PL consistently improves the link predictor, but not as much as the CPL. On WikiCS however, PL keeps weakening the AUC, as the introduced noise outweighs. On Amazon_Photo, PL can improve the performance at the first few iterations. But the AUC suddenly drops from 70% to around 50% after iterations, which illustrates that introducing too much noise may degrade the model. As for the CPL, although it may drop after reaching its best, i.e. for WikiCS, it can distinctively improve the performance compared to the base model and will not lead to failure.

### 3.2.5 Impact of multi-view augmentation

During the training, we apply multi-view augmented graphs as the input of the teacher model, so that the continuity condition in Theorem 2.3 is easier to be satisfied. We here conduct the ablation experiments that compares multi-view augmentation with different single-view augmentation methods, including drop node (Node view), feature mask (Feature view), and DropEdge (Structure view). In each experiment, a single augmentation method was applied three times. And "Random" refers to the random selection of samples during PL. As shown in Fig.4, all of the single-view augmentation methods have better performance than the base model and their improvements are almost the same. When we use multi-view augmentation, the AUC and AP are further improved. It illustrates that multi-view augmentation contributes to a more robust graph learning and tends to obtain a consistent result, which echoes with our theoretical analysis.

Table 4: CPL with different $k$.

| | **CiteSeer** | | | |
|---|---|---|---|---|
| **k** | Raw | 100 | 500 | 2000 |
| **AUC(%)** | 71.10 | 72.25 | 72.45 | 71.98 |
| **AP(%)** | 72.45 | 73.29 | 73.54 | 73.13 |
| | **Amazon_Photo** | | | |
| **k** | Raw | 5 | 50 | 500 |
| **AUC(%)** | 67.92 | 76.63 | 76.30 | 75.95 |
| **AP(%)** | 67.06 | 75.70 | 75.52 | 74.97 |

Table 5: Case analysis of CPL on LastFMAsia.

| | **GCN** | **GraphSAGE** | **GAT** | **APPNP** |
|---|---|---|---|---|
| Inconsistency $\mathcal{A}$ (%) | 6.69 | 4.01 | 2.96 | 3.13 |
| Confidence $1 - q$ (%) | 77.63 | 88.35 | 83.00 | 86.86 |
| Theoretical $\mathrm{Err}_{th}$ (%) | 58.12 | 31.32 | 39.92 | 32.54 |
| Experimental $\mathrm{Err}_{exp}$ (%) | 20.08 | 17.75 | 17.11 | 16.44 |
| PL error (%) | 7.78 | 6.43 | 8.18 | 6.02 |
| M3S PL error (%) | 65.63 | 27.00 | 65.00 | 27.49 |
| DR-GST PL error (%) | 26.31 | 14.10 | 13.38 | 29.09 |

## 3.3 Case Study

The case study on node classification is conducted on LastFMAisa on different baselines. The detailed intermediate variables about the error analysis are listed in Table 5. We record the inconsistency $\mathcal{A}$ and confidence $1 - q$ during the CPL as Algorithm.1. Then the theoretical error bound $\mathrm{Err}_{th}(g)$ is calculated by Theorem 2.3. As $\mathrm{Err}_{th}(g) > \mathrm{Err}_{exp}(g)$, the experimental error is bounded by the theoretical error, implying that it is an effective bound. We also report the accuracy of PL samples. The CPL has smaller error than other PL strategies and is consistently better than $\mathrm{Err}_{exp}(g)$.

The case study on link prediction is conducted based on GAE on WikiCS. The relationship between consistency $1 - \mathcal{A}(g)$ and the number of PL samples is shown in Fig.5. The result shows that more PL samples help to increase the consistency of the prediction. We also compare the optimization target of PL $\mathcal{L}_{\mathcal{R}}^{(t)}$ and CPL $\mathcal{L}_{\mathcal{T}}^{(t)}$ in Fig.5. We discover that the optimization target of CPL converges faster than PL and is consistently smaller. It

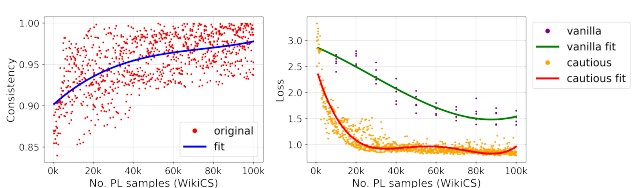

Figure 5: Case study of the consistency and convergence on WikiCS: Cautious PL (CPL) improves the prediction consistency within error bound, and converges faster than PL.

implies the covariance term $\mathrm{Cov}\,[\mathrm{ce}, \mathcal{T}]$ is negative as illustrated in Theorem 2.5. Thus, the CPL strategy can improve the convergence property of graph learning. A detailed illustration of error bound verification and Knowledge discovery is shown in Appendix E.

## 4 Related Works

Pseudo labeling is a popular approach in self-training (ST), aiming to enlarge the training set by self-annotation. Most studies focus on learning a more accurate PL algorithms to avoid noisy samples. Confidence thresholding is a simple yet effective method in ST, where the similarity with ground truth or consistency is used to measure the confidence, ensuring that flawed samples are excluded from the enlarged dataset [9, 20]. [6] uses the perturbation on the hidden states to yield close predictions for similar unlabeled inputs. [7, 16, 26] rely on the consistency of the unsupervised clustering. [30] utilizes adversarial learning to acquire domain-uninformative representations and train a PL classifier. [3, 17] use a contrastive loss to improve the representation learning before PL. Directly modeling noisy labels can also enhance noise tolerance [31], such as using soft pseudo labels [2], or distilling correct information to mitigate overfitting on noise [32].

The confidence measure plays a crucial role in avoiding the overconfident results [11]. [33] constructs confidence regularizers, expecting a smooth prediction with soft-label and preventing infinite entropy minimization. [10] learns a confidence metric based on the generality and unity of its distribution of pseudo loss. [1] uses the distance between the distributions of positive and negative samples as a confidence measure. [24] applies data augmentation on node features, where consistency is used as a confidence measure for classification. The co-training method constructs multi-view classifiers. It adds pseudo labels in each view to provide complementary information for each other, showing better performance than single-view [5].

Some studies on graph data apply the PL strategy to node classification tasks. M3S [21], IFC-GCN[8] utilize the clustering to PL the unlabeled samples. CaGCN-st [23] is a confidence-calibrated model

that utilizes low-confidence but high-accuracy samples. DR-GST [18] employs dropout and edge-drop augmentation to conduct information gain inference for selecting PL samples.

## 5 Conclusion

In this study, we provide deep insights into PL strategy by analyzing its impact on prediction error and the convergence properties. We offer theoretical explanations for the effect of PL strategies on graph learning processes, particularly addressing degradation issues. Based on the theoretical analysis, we introduce the CPL strategy, a plug-in and practical technique that can be generally applied to various baseline models. The experiments demonstrate effectiveness and superiority of CPL in link prediction and node classification tasks.

In future work, we plan to explore a more reliable confidence measures as the PL criteria, such as informativeness in the multi-view network and prediction uncertainty.

## Acknowledgments and Disclosure of Funding

The research of Tsung was supported in part by the Industrial Informatics and Intelligence (Triple-i) Institute at HKUST(GZ).

The research of Jia Li was supported by NSFC Grant No. 62206067, Tencent AI Lab Rhino-Bird Focused Research Program and Guangzhou-HKUST(GZ) Joint Funding Scheme 2023A03J0673.

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

# A  Proof of Proposition 2.2: additive expansion proposition

We denote the embedding vector of a node $v_i$ by $e_i \triangleq g_i(\hat{G}) = \text{GNN}(\hat{G})[i]$. Without loss of generality, we drop the subscript for short. We can also define the density measure $d_f$ as:

$$p_f(e) \triangleq \frac{\exp(-d(e))}{\int_{\mathcal{G}} \exp(-d(e)) dg(\mathcal{G})}. \tag{5}$$

For any subset of the embedding space $E \subset g(\mathcal{X})$, the local probability can be measured by $p_f(E) = \int_E p_f(e) de$. If we have a local optimal subset $U \subset E$ with a confidence threshold of $1 - q$, and its perturbation $U_\epsilon$, then the consistency at the boundary of the subset separation problem $E \to U, E \setminus U$ can be quantified by Cheeger constant. We introduce the continuous Cheeger inequality to elaborate the lower bound of the Cheeger constant under an amplified measure $\alpha f$, where $\alpha > 1$ is a constant.

We first define the restricted *Cheeger constant* in the link prediction task. Given the function $f$ and any subset $E$, Cheeger constant is calculated by

$$\mathcal{C}_f(E) \triangleq \lim_{\epsilon \to 0^+} \inf_{A \subset E} \frac{p_f(A_\epsilon) - p_f(A)}{\epsilon \min\{p_f(A), p_f(E \setminus A)\}}. \tag{6}$$

According to the definition, the Cheeger constant is a lower bound of probability density in the neighborhood of the given set. It quantifies the chance of escaping the subset $A$ under the probability measure $f$ and reveals the consistency over the set cutting boundary.

Then we prove that for the any subset $E \subset g(\mathcal{G})$ with its local optimal subset $U : \{e \in E : p_f(e) > 1 - q\}$, there exists $\alpha > 1$ s.t. $\mathcal{C}_{\alpha f}(E \setminus U) \geq 1$.

As the measurable function for the link prediction is defined as $f(e) = -e^T e_a$. When $e^* = e_a$, $f$ reaches the global minimal. For the embedding vectors outside the local minimal subset $e_y \in g(\mathcal{G}) \setminus U$, there exsits $\epsilon > 0$ s.t.

$$f(e_y) \geq f(e^*) + 2\hat{\epsilon}, \tag{7}$$

where $\hat{\epsilon} = C\epsilon$. If we define $E_\epsilon^* = \{e_{\hat{\epsilon}}^*\} \cap g(\mathcal{G})$, where $e_{\hat{\epsilon}}^*$ is the $\hat{\epsilon}$ neighbor of $e^*$, according to the Lipchitz condition of $f$, for $e \in E_\epsilon^*$, we have:

$$f(e_x) \leq f(e^*) + \hat{\epsilon} \|e_x - e^*\|_2 \leq f(e^*) + \hat{\epsilon}. \tag{8}$$

Combining Eq.8 and Eq.7 leads to $f(e_y) - f(e_x) \geq \hat{\epsilon}$. Thus, for the amplified probability measure $p_{\alpha f}$, we have

$$p_{\alpha f}(e_x)/p_{\alpha f}(e_y) \geq \exp(\alpha\hat{\epsilon}) \tag{9}$$

According to the inequality property from [28] (formula 63), we have

$$\frac{p_{\alpha f}(U)}{p_{\alpha f}(g(\mathcal{G}) \setminus U)} \geq \exp\left(\alpha\hat{\epsilon} - 2\log\left(2C^2/\hat{\epsilon}\right)\right). \tag{10}$$

As $p_{\alpha f}(g(\mathcal{G}) \setminus U) + p_{\alpha f}(U) = 1$. If we select $\alpha$ large enough s.t. the RHS of Eq.10 is larger than 1, $p_{\alpha f}(g(\mathcal{G}) \setminus U) \leq \frac{1}{2}$. Thus, according to [19] (Theorem 2.6), we have $\mathcal{C}_{\alpha f} \geq 1$. It guarantees consistency around the perimeter of the $U$. As $\alpha > 1$ and $p_f, p_{\alpha f}$ are bounded on any subsets of embedding space. It implies a probability margin $\eta > 0$ at the neighborhood of the local optimal between two measurable functions $f, \alpha f$, where

$$\eta = \inf_{\hat{e} \in U_\epsilon \setminus U, e \in U} (p_{\alpha f}(\hat{e}) - p_f(e)). \tag{11}$$

which, according to [25], implies additive expansion property of the probability measure in the link prediction, as Proposition 2.2.

# B   Proof of Theorem 2.3: error analysis

In [25], $\mathcal{M}(g_\phi)$ is also assumed to satisfy additive-expansion $(q, \epsilon)$, where $\mathcal{M}(g) \triangleq \{y \in Y : g(y) \neq y\}$ is the set of mis-classified samples, and they give the error bound of the trained classifier $s$ (Theorem B.2):

$$\text{Err}(g) \leq 2(q + \mathcal{A}(g)). \tag{12}$$

Here in link prediction task, $\mathcal{M}(g_\phi)$ is mis-classified samples by the pseudo labeler (teacher model). It can be written by $\{y_i : Y_p[i] \neq \mathcal{E}_\mathcal{T}[i]\}$, which is intractable during the training. The probability threshold is $1 - q$ and a local optimal subset $U$ for PL is constructed accordingly. We aim to let $\mathcal{M}(g_\phi) \cap U$ be close to $\emptyset$, so that $g(\mathcal{G}) \setminus U$ can cover $\mathcal{M}(g_\phi)$ as much as possible. So we define the robust set $\mathcal{S}(g)$ as

$$\mathcal{S}(g) = \{y : g(y) = g(\hat{y}), \hat{y} \in \{y_\epsilon\}\}, \tag{13}$$

where $y_\epsilon$ is the $\epsilon$ neighborhood of sample $y$. Then, according to Proposition 2.1, we have:

$$p_f(\{y \in Y : g_\phi(y) \neq y, y \in \mathcal{S}(g_\psi)\}) \leq p_{\alpha f}(g(\mathcal{G}) \setminus U) \leq q, \tag{14}$$

which has similar form with [25] Lemma B.3 for link prediction task. Besides, the analysis of $p_f(\{y \in Y : g_\phi(y) = y, g_\psi(y) \neq y, y \in \mathcal{S}(g_\psi)\})$ and $p_f(\overline{\mathcal{S}(g_\psi)})$ are the same. Thus, the assumption on $\mathcal{M}(g_\phi)$ is satisfied. Then, we can draw the same conclusion with Eq.12, and the classifier is the student model $g_\psi$. The theorem is proofed.

# C   Proof of convergence inequality

The PL strategy $\mathcal{T}$ for the unlabeled data provides a Bayesian prior, from which we formalize the empirical loss defined in Eq.1 as

$$\mathcal{L}_\mathcal{T}^{(t+1)} = \frac{1}{\left|\hat{Y}_o^{(t)}\right| + k} \left[ \text{CE}\left(g_\psi^{(t)}, \hat{Y}_o^{(t)}\right) + \text{CE}\left(g_\psi^{(t)}, Y_p^{(t)}\right) \right]. \tag{15}$$

We can decompose the cross-entropy loss of the pseudo labeled samples by:

$$
\begin{aligned}
\text{CE}(g_\psi, Y_p) &= \sum_{\hat{Y}_u} \text{ce}(g_\psi, Y) \cdot \mathcal{T} \\
&= \sum_{\hat{Y}_u} [\text{ce}(g_\psi, Y) - Y[\text{ce}(g_\psi, Y)]] \cdot [\mathcal{T} - Y\mathcal{T}] \\
&\quad + Y\mathcal{T} \sum_{\hat{Y}_u} \text{ce}(g_\psi, Y) + \mathbb{E}[\text{ce}(g_\psi, Y)] \sum_{\hat{Y}_u} \mathcal{T} - \left|\hat{Y}_u\right| Y\mathcal{T}Y[\text{ce}(g_\psi, Y)]
\end{aligned}
\tag{16}
$$

.

Thus, Eq.16 can be simplified to:

$$
\begin{aligned}
\text{CE}(g_\psi, Y_p) &= \left|\hat{Y}_u\right| \text{Cov}[\text{ce}(g_\psi, Y), \mathcal{T}] + \mathbb{E}\mathcal{T} \cdot \left|\hat{Y}_u\right| \mathbb{E}[\text{ce}(g_\psi, Y)] \\
&\quad + \mathbb{E}[\text{ce}(g_\psi, Y)] \cdot \left|\hat{Y}_u\right| \mathbb{E}\mathcal{T} - \left|\hat{Y}_u\right| \mathbb{E}\mathcal{T}\mathbb{E}[\text{ce}(g_\psi, Y)] \\
&= \left|\hat{Y}_u\right| \text{Cov}[\text{ce}(g_\psi, Y), \mathcal{T}] + \left|\hat{Y}_u\right| \mathbb{E}\mathcal{T}\mathbb{E}[\text{ce}(g_\phi, Y)] \\
&= \left|\hat{Y}_u\right| \text{Cov}[\text{ce}(g_\psi, Y), \mathcal{T}] + k\mathbb{E}[\text{ce}(g_\phi, Y)]
\end{aligned}
\tag{17}
$$

Note that $\mathcal{T}$ is the indicator-like function, where we have

$$\mathbb{E}\mathcal{T} = \frac{1}{\left|\hat{Y}_u\right|} \sum_{\hat{Y}_u} \mathcal{T} = \frac{k}{\left|\hat{Y}_u\right|}. \tag{18}$$

Based on the Eq.15 and Eq.17, we can rewrite $\mathcal{L}_\mathcal{T}^{(t+1)}$ as

$$\mathcal{L}_\mathcal{T}^{(t+1)} = \beta \text{Cov}\left[\text{ce}\left(g_\psi, Y\right), \mathcal{T}\right] + \frac{1}{\left|\hat{Y}_o^{(t)}\right|} \text{CE}\left(g_\psi^{(t)}, \hat{Y}_o^{(t)}\right)$$

$$\leq \beta \text{Cov}\left[\text{ce}\left(g_\psi, Y\right), \mathcal{T}\right] + \frac{1}{\left|\hat{Y}_o^{(t)}\right|} \text{CE}\left(g_\phi^{(t)}, \hat{Y}_o^{(t)}\right) \qquad (19)$$

$$= \beta \text{Cov}\left[\text{ce}\left(g_\psi, Y\right), \mathcal{T}\right] + \mathcal{L}_\mathcal{T}^{(t)}$$

where $\beta = |\hat{Y}_u|/(|\hat{Y}_o| + k$. The inequality holds due to the assumption.

Table 6: Time consumption of CPL on link prediction

| Base model | CiteSeer | Actor | WikiCS | TwitchPT | AmazonPhoto |
|---|---|---|---|---|---|
| GAE | 452 | 6376 | 7412 | 3401 | 4956 |
| node2vec | 691 | 5067 | 6537 | 2740 | 5470 |
| SEAL | 2982 | 14579 | 17491 | 11639 | 24904 |

Table 7: Time consumption of CPL on node classification

| Base model | Cora | CiteSeer | PubMed | AmazonPhoto | LastFMAsia |
|---|---|---|---|---|---|
| GCN | 115.8 | 241 | 147.6 | 302.6 | 104 |
| GAT | 251.4 | 450 | 388.8 | 652.2 | 338.6 |
| SAGE | 141.6 | 237.8 | 181.2 | 347 | 134.6 |
| APPNP | 511.2 | 1005.2 | 820 | 843.75 | 576.2 |

## D  Time consumption of CPL

The time consumption of CPL on node classification and link prediction are shown in Table 6 and Table 7 respectively.

## E  Case study of CPL on link prediction

**Error bound:** In the case study, the recorded confidence threshold is $1 - q = 0.98$ for WikiCS. We adopt 5 views of dropout with the augmentation drop rate 0.05. And according to the error bound given by Theorem 2.3, given the confidence threshold, Eq.2 suggests that the higher prediction consistency should lead to a smaller error bound. The final prediction consistency is $\mathcal{A}(g) = 0.0358$, thus, we can calculate error bound $Err(g) = 0.1116$. The AUC and AP are $95.56 \pm 0.24\%, 95.58 \pm 0.29\%$ which are bounded within $Err(g)$.

**Knowledge discovery:** In the 5 random experiments, we add 500 pseudo links in each iteration. Here we focus on the common PL links in the first iteration, which are considered the most confident samples. We look for the metadata of WikiCS whose node, feature, link and node label represent paper, token, reference relation and topic of the paper respectively. There are These 7 most confident links categorized into 2 groups. We take 3 out of 5 nodes in group1 and the 2 nodes in group2 for analysis, whose detailed information of these nodes is shown in AppendixE.

Table 8: Details of Node Information in the Case Study.

| Node | Group 1 | | | Group 2 | |
|---|---|---|---|---|---|
| | **Node 2702** | **Node 5688** | **Node 8906** | **Node 3489** | **Node 7680** |
| ID | 17505908 | 11353631 | 30138652 | 23221074 | 12265137 |
| Outlinks | [6097297] | [6097297] | [244374, 6097297] | [20901] | [] |
| Title | Ubuntu Hacks | Pungi (software) | LinuxPAE64 | Malware Bell | Norton Confidential |
| Label | Operating systems | Operating systems | Operating systems | Computer security | Computer security |
| Tokens | "ubuntu", "hacks", "ubuntu", "hacks", "tips", "tools", "exploring", "using", "tuning", "linux", "book", "tips", "ubuntu", "popular", "linux", "distribution", "book", "published", "o'reilly", "media", "june", "2006", "part", "o'reilly", "hacks", "series" | "pungi", "software", "pungi", "program", "making", "spins", "fedora", "linux","distribution", "release", "7", "upwards" | "linuxpae64", "linuxpae64", "port", "linux", "kernel", "running", "compatibility", "mode", "x86-64", "processor", "kernel", "capable", "loading", "i386", "modules", "device", "drivers", "supports", "64-bit", "linux", "applications", "user", "mode" | "malware", "bell", "malware", "bell", "malware", "program", "made", "taiwan", "somewhere", "2006", "2007", "malware", "bell", "tries", "install", "automatically", "upon", "visiting", "website", "promoting", "containing", "malware" | "norton", "confidential", "norton", "confidential", "program", "designed", "encrypt", "passwords", "online", "detect", "phishing", "sites" |

For group 1, 3 nodes are connected by the pseudo links, and they are all linked to a central node whose degree is 321. The metadata information of the nodes are all strongly relevant to "Linux" in the "operating systems" topic. Thus, the PL linked nodes are likely to have common neighbors discovered triangle relationship. In group2, node 3489 has no in/out degree and is pseudo linked to node 7680. Both papers focus on the "malware"/"phishing" under the topic "Computer security". Although they only have one common token, the CPL strategy successfully discovers the correlation and consistently add it to the training set. The detailed result of the case study is shown in Table 8.

