# OpenReview forum: "Deep Insights into Noisy Pseudo Labeling on Graph Data"
_NeurIPS.cc/2023/Conference — NeurIPS 2023 poster_

### Official Review · Reviewer_4kBQ · 2023-07-01

**Soundness:** 3 good
**Presentation:** 3 good
**Contribution:** 3 good
**Rating:** 5
**Confidence:** 4

**Summary:**

This paper aims to provide in-depth insights into pseudo labeling (PL) in the context of graph learning models. The authors first present an error analysis of the PL strategy, demonstrating that the error is bounded by the confidence threshold of PL and the consistency of multi-view predictions. Furthermore, they theoretically illustrate the impact of PL on convergence properties. Building upon this analysis, they propose a careful pseudo labeling methodology that involves assigning pseudo labels to samples with the highest confidence and multi-view consistency. Finally, extensive experiments demonstrate that the proposed strategy enhances the graph learning process and outperforms alternative PL strategies in link prediction and node classification tasks.

**Strengths:**

1. This paper theoretically analyzes how the PL strategy affects convergence properties in GNN.
2. Based on the analysis, this paper proposes a cautious pseudo labeling methodology, and extensive experiments demonstrate the effectiveness of the proposed strategy.

**Weaknesses:**

1.There might be some mistake in Theorem 2.5. since the covariance between the cross-entropy loss and PL strategy and $\beta$ is non-negative, the inequality $\beta Cov + L \leq L$ holds only if $\beta Cov = 0$.
2.It might be better to clearly illustrate $q(t)$ in Figure 2, which demonstrates the framework of the algorithm.
3.It is suggested to remove the first picture in Figure 1 and add two more pictures to show the performance on node classification.
4.The word ‘noisy’ in the title is redundant for pseudo labeling. Furthermore, the whole paper rarely mentions ‘noisy PL’.

**Questions:**

How much time does the proposed method consume on the dataset like PubMed?

**Limitations:**

Some figures could be further polished up, and check the correctness of the theorem.

---

> ### Author Rebuttal · Authors · 2023-08-10
>
> We thank the reviewer for the careful reading and insightful comments. Following are the responses regarding your concerns.
>
> For the figures in the rebuttal, please check the PDF file in the global response of the author rebuttal, located at the top of this page.
>
> > 1.There might be some mistake in Theorem 2.5. since the covariance between the cross-entropy loss and PL strategy and  \beta  is non-negative, the inequality $\beta Cov + L \leq L$  holds only if $\beta Cov = 0$
>
> We are confident that Theorem 2.5 contains no errors. Our analysis in lines 184-190 demonstrates that the covariance term, denoted as Cov, is never greater than 0. Considering that \beta is non-negative, the product of \beta and Cov would be non-positive. Consequently, the loss decreases iteratively, leading to a smaller overall loss.
>
> > 2.It might be better to clearly illustrate  q(t)  in Figure 2, which demonstrates the framework of the algorithm. 3.It is suggested to remove the first picture in Figure 1 and add two more pictures to show the performance on node classification. 4.The word ‘noisy’ in the title is redundant for pseudo labeling. Furthermore, the whole paper rarely mentions ‘noisy PL’.
>
> Thanks for your advice. We add the q(t) in Fig.2, as Fig. a in the attachment of our very first response.
>
> We have conducted experiments to investigate the impact of PL capacity on node classification performance. The results indicate that initially, as the number of PLs increases, there is an improvement in model performance. Then the improvement is followed by a degradation in performance until no nodes remain above the confidence threshold. It is important to note that due to the significantly smaller number of nodes compared to edges, our experiments only capture the first scenario illustrated in Fig. 2, as shown in Fig. b,c of the attachment in our very first response. This is why we specifically selected the PL experiment for the link prediction task in the introduction.
>
> The primary contribution of our study lies in the theoretical analysis of the error induced by PL, which is a major drawback associated with noisy labels. The entire paper focuses on quantifying and mitigating the influence of noisy PL, addressing this critical issue in label noise research.
>
> > How much time does the proposed method consume on the dataset like PubMed?
>
> The averaged time consumption (seconds) on PubMed is reported as the following table:
>
> | Base model | GCN | GAT | SAGE | APPNP |
> | -- | -- | -- | -- | -- |
> | Time(s) | 147.6 | 388.8 | 147.6 | 181.2 | 820.0 |
>
> > Some figures could be further polished up, and check the correctness of the theorem.
>
> Thanks for your advice. We would update to more clear figures in the latest version and correct the typos in the paper. Please refer to the attachment in our first overall response.

---

> > ### Comment · Reviewer_4kBQ · 2023-08-13
> > **About the rebuttal**
> >
> > 1. The authors have addressed most of my concerns.
> >
> > 2. The authors mention that the covariance term, denoted as Cov, is never greater than 0 in the rebuttal. However, in Page 5, line 184-185, they say that "the second inequality holds because the covariance between the cross-entropy loss and PL strategy is non-negative". Does there exist a mistake?

---

> > > ### Author Response · Authors · 2023-08-13
> > >
> > > We apologize for the typographical error in Line 184-185. Based on the analysis conducted in Line 184-190, it has been determined that the $Cov$ term in Eq.4 should be non-positive, as indicated in Line 190. We have made the necessary revision in the latest version.

---

### Official Review · Reviewer_io9b · 2023-07-06

**Soundness:** 3 good
**Presentation:** 3 good
**Contribution:** 2 fair
**Rating:** 5
**Confidence:** 3

**Summary:**

Pseudo labeling is significant for GNN. This paper first theoretically analyzed the effect of pseudo labeling by showing the error bound and the convergence property. Then, accordingly, the paper proposes a cautious pseudo labeling based on confidence and multi-view consistency. The experimental results demonstrate the effectiveness of the proposed cautious pseudo labeling strategy.

**Strengths:**

The paper is well written. The analysis of Pseudo labeling for GNN is valuable.

**Weaknesses:**

However, the analysis or the conclusion is general to all fields instead of highly related to the graph. Besides, the solutions, including high confidence and multi-view consistency, are not that novel.

More experiments are required. The time complexity should be provided when comparing with previous methods. Performance comparison on link prediction should also contain the previous PL methods.



**Questions:**

1) The paper is well written. The analysis of Pseudo labeling for GNN is valuable. However, the analysis or the conclusion is general to all fields instead of highly related to the graph. Besides, the solutions, including high confidence and multi-view consistency, are not that novel.

2) Some contents may contain errors. For example, the results of Figure 1 may contain some errors. When the samples of PL and CPL are equal to/similar to 0, the GAE/GAE+PL/GAE+CPL should have equal/similar performance.

3) More details need to be provided. What are the details of multi-view teachers? Does it refer to multi teachers or one teacher with multi-data-augmentation inputs? If the multi-view is obtained based on the data augmentation, what types of data augmentation are used? What is the data augmentation in Figure 4?

4) More experiments are required. The time complexity should be provided when comparing with previous methods. Performance comparison on link prediction should also contain the previous PL methods.

5) Other questions. In Algorithm 1, the student was fine-tuned based on the teacher model. I wonder whether there exists overfitting since the observed data set has been put into the model again and again. In 3.1, I wonder whether the data from the testing would be selected by CPL to be labeled for training. I wonder whether CPL may reduce the performance of nodes of low-frequently labeled classes.

**Limitations:**

There is no potential negative societal impact.

---

> ### Author Rebuttal · Authors · 2023-08-10
>
> We thank the reviewer for the thoughtful comments. Please find our response which addresses your concern.
>
> 1.
>
> Thanks for your agreement that our paper is well written and this is really a good question. Our analysis of the PL strategy focuses on the task of graph learning. We make assumptions regarding the graph properties, i.e., Graph Perturbation Invariant (GPI) property and Additive Expansion Property (AEP). GPI assumes that the representation of the graph does not undergo significant changes when augmentation is applied. AEP assumes the continuity of the probability density in the neighborhood of the local optimal set. It is highly likely that these assumptions hold true for GNNs. The idea of message passing in GNNs smooths out the discrete variations caused by augmentation and ensures continuity. Additionally, GNNs are typically shallow networks, which implies the existence of a measure that prevents extreme fluctuations in values. However, it remains unclear whether these assumptions hold in other scenarios such as tabular data.
>
> The main contribution of our study lies in providing insights into PL in the context of graph learning. While thresholding and multiview augmentation techniques are not new in the machine learning community, their adoption in our study is a natural solution as indicated by our analysis, rather than trivial combinations in other empirical studies. We aim to quantify the error bound of PL by incorporating these techniques. Our theoretical analysis demonstrates that the performance of PL is influenced by these two factors. Therefore, we combine them to the PL and quantify their impact. Specifically, the noisy PL belong to the augmentation space, and thresholding is used to filter the local optimal set, and its consistency can be considered an explicit measure of the error bound of PL.
>
> 2.
>
> The results depicted in Fig.1 are correct based on our experiments. These experiments were conducted with specific PL capacities, i.g. 2k, 4k, ..., and 100k for the WiKiCS. And we employed a polynomial fitting curve. To avoid overfitting, we refrained from using a high-order polynomial. Thus, there might be two factors contributing to the initial value problem:
>
> a) The original training set is relatively small. Consequently, the initial PL samples have a relatively substantial impact and enhance the prediction performance. This also influences the initial value.
>
> b) The fitting result is influenced by the nonlinear relationship between the variables. The nonlinearity can affect the initial fitting value of the function.
>
> 3.
>
> "multi-view teacher model" refers to a single teacher model with multiple inputs, each of which is augmented separately. We apply various data augmentations to the original graph. These augmented graphs are fed into the teacher model individually. The final prediction is the average predictions generated from these inputs. In the next, we analyze the consistency among these predictions.
>
> In Fig.4, we employed three different augmentation methods: drop node (Node view), feature mask (Feature view), and DropEdge (Structure view). In each experiment, a single augmentation method was applied three times. As for Multiview, we applied each augmentation method once as a combined augmentation. The "Random'' refers to the random selection of samples during PL.
>
>  4.
>
> The overhead of the CPL stems from two main components: the calculation of confidence scores and the fine-tuning process on the enlarged training set. In each iteration, the time required to compute the confidence scores for the PL candidates is approximately the evaluation time of the test set. For the fine-tuning, the total epoch number is roughly twice that of  the pre-training epochs. The total time consumption for fine-tuning is expected to be twice that of the base model.The averaged time consumption (second) :
>
> Node classification:
>
> | Base model | Cora | CiteSeer | PubMed | AmazonPhoto | LastFMAsia |
> | -- | -- | -- | -- | -- | -- |
> | GCN | 115.8 | 241.0 | 147.6 | 302.6 | 104.0 |
> | GAT | 251.4 | 450.0 | 388.8 | 652.2 | 338.6 |
> | SAGE | 141.6 | 237.8 | 181.2 | 347.0 | 134.6 |
> | APPNP | 511.2 | 1005.2 | 820.0 | 843.75 | 576.2 |
>
> Link prediction:
>
> | Base model | CiteSeer | Actor | WikiCS | TwitchPT | AmazonPhoto |
> | -- | -- | -- | -- | -- | -- |
> | GAE | 452 | 6376 | 7412 | 3401| 4956 |
> | node2vec | 691 | 5067 | 6537 | 2740 | 5470 |
> | SEAL | 2982 | 14579 | 17491 | 11639 | 24904 |
>
> We compare with another study, i.e. EdgeProposal, that proposes a similar approach for link prediction by introducing possible edges during training. The comparison experiment is provided, with metric Hit@20 (%)
>
> | Dataset |GAE | EdgePropsal | CPL |
> | -- | -- | -- | -- |
> | ogb-ddi | 41.4 | 53.4 | 58.9 |
> | ogb-colab | 60.1 | 60.4 | 60.6 |
>
> 5.
>
> We acknowledge that one of the weaknesses of PL-based methods is the risk of overfitting as PL samples often have similar representations to the original training set. However, the error-labeled samples may have a significant influence on the model. To prevent the model from being misled by the error introduced, it is necessary to retain the original training set. Besides, we incorporated a validation set, and the results demonstrated that the CPL does not suffer from overfitting.
>
> The proposed model is transductive learning. The test set is also the candidate for PL. However, their labels are totally isolated from the training process.
>
> We did not specifically observe the outcome of imbalanced PL. To explore this further, let us consider an extreme story in which we only pseudo label some classes rather than all classes. This experiment was conducted on CiteSeer with GCN for the node classification task. During CPL, we adjust the imbalance ratio by only pseudo labeling some specific classes.
>
> Imbalance CPL:
>
> | # PL class | 0(raw) | 1 | 2 | 3 | 4 | 5  | 6(balanced) |
> | -- | -- | -- | -- | -- | -- | -- | -- |
> | AUC(%) | 69.32 | 69.65 | 70.02 | 70.12 | 70.93 | 72.22 | 72.96 |

---

> > ### Comment · Reviewer_io9b · 2023-08-14
> >
> > The authors have addressed most of my concerns.
> >
> > I agree with that: "While thresholding and multiview augmentation techniques are not new in the machine learning community, their adoption in our study is a natural solution as indicated by our analysis, rather than trivial combinations in other empirical studies. "

---

### Official Review · Reviewer_iJpS · 2023-07-07

**Soundness:** 4 excellent
**Presentation:** 1 poor
**Contribution:** 4 excellent
**Rating:** 7
**Confidence:** 4

**Summary:**

The paper provides an error bound for pseudo labeling on graphs. Moreover, the authors propose a cautious pseudo labeling method and validate it through experiments.

**Strengths:**

1. The paper presents good experimental results, demonstrating the effectiveness of the proposed cautious pseudo labeling method.
2. The authors have conducted comprehensive experiments
3.  The error bound proposed in the paper is deemed useful and practical

**Weaknesses:**

The paper suffers from unclear notation and contains numerous typos. These issues hinder the reader's understanding and make it challenging to follow the presented ideas. Clarifying the notation and addressing the typos is necessary to improve the clarity of the paper.

**Questions:**

1. Line 86: For "err(g)", does it mean the expectation is taken over the test points? Please clarify this point.

2. Line 91: Is the GPI property introduced in this paper, or has it been introduced previously? Please provide the necessary context for understanding this property.

3. Line 104: When stating "whose probability is higher than a threshold," does it mean this condition must hold for each "y ∈ U"? Please clarify this point.

4. Line 105: Since "g(G) ∈ ℝ^(N×M)", what do you mean by "ŷ ∈ g(G)"? Please explain this notation.

5. Line 102: Do you mean we can find "p_f(⋅)" or we can find "α, η"? Please clarify this statement.

6. Line 108: In "p_{αf}", there should be no "α" here. Please correct this notation.

7. Line 122: When stating "the teacher predictor satisfies additive expansion," the additive expansion is defined for a probability density, not a classifier. Please provide clarification and ensure consistency in terminology.

8. Line 124: Could you explain what is meant by "E_{Y_test}"?

9. Line 145: The first term should change "t" to "t+1".

10. Line 147: What is the definition of the covariance term in this context? Please provide clarification.

11. Lines 163-164: There seems to be a typo in the sentence, "we calculate multi-view prediction of the by." Please clarify this phrase.

12. Do you have any formal proof of Theorem 2.5?

I am willing to increase my score if these concerns are resolved.

**Limitations:**

The authors have adequately addressed the limitations

---

> ### Author Rebuttal · Authors · 2023-08-10
>
> Thank you for taking the time to review our paper. We sincerely appreciate your efforts in providing us with a detailed review. We have carefully considered all of your insightful suggestions and corrections, and we have incorporated them into the latest version of our draft. We have addressed each comment individually as follows.
>
> > 1. Line 86: For "err(g)", does it mean the expectation is taken over the test points?
>
> Yes. It measures the performance of the proposed model on the test set. Besides, we also use it to quantify the PL accuracy during the training, where the expectation is taken over the selected PL samples.
>
> > 2. Line 91: Is the GPI property introduced in this paper, or has it been introduced previously?
>
> Analogous analyses of GPI are frequently encountered, such as "label-invariant augmentation" as Definition 4.1 in [1], or the "augmentation invariant" in self-supervised learning as Definition 1in [2]. These analyses typically assume the invariance of categories/representation or serve as regularization techniques. In our approach, we relax these constraints by incorporating the Lipschitz regularity, a principle commonly employed in the theoretical analysis of neural networks, as Definition 1in [3] and Definition 1in [4].
>
> [1] Yu, Junchi, Jian Liang, and Ran He. "Mind the Label Shift of Augmentation-based Graph OOD Generalization." IEEE/CVF CVPR. 2023.
>
> [2] Hua, Tianyu, et al. "On feature decorrelation in self-supervised learning." IEEE/CVF ICCV. 2021.
>
> [3] Virmaux, Aladin, and Kevin Scaman. "Lipschitz regularity of deep neural networks: analysis and efficient estimation." NIPS 31 (2018).
>
> [4] Arghal, Raghu, Eric Lei, and Shirin Saeedi Bidokhti. "Robust graph neural networks via probabilistic Lipschitz constraints." Learning for Dynamics and Control Conference. PMLR, 2022.
>
> > 3. Line 104: When stating "whose probability is higher than a threshold," does it mean this condition must hold for each "y ∈ U"?
>
> Yes, the probability of the elements $y$ in the local optimal set $U$ should be higher than the threshold.
>
> > 4. Line 105: Since "g(G) ∈ ℝ^(N×M)", what do you mean by "ŷ ∈ g(G)"?
>
> Sorry for the notation mistake. g(G) is the output N×M confidence matrix, ŷ is the N×M probability prediction of the augmented graph. It should be ŷ = g(G) here.
>
> > 5. Line 102: Do you mean we can find "p_f(⋅)" or we can find "α, η"? Please clarify this statement. & 6. Line 108: In "p_{αf}", there should be no "α" here.
>
> Thanks for your judgment. The conclusion should be p_{αf}(U/Uε)>=p_{αf}(U)+αη. This proposition is an assumption on the continuity of the trained GNN-based confidence predictor over the local optimal set. When we apply the graph augmentation, the perturbed local optimal set becomes larger. The increased subset still satisfies the inequality under amplified measure. We aim to find the probability measure for the prediction "p_f(⋅)" rather than "α, η", which refers to the trained GNN in the experiment. We only wish to illustrate the continuity property. Thus, the proposition only states the existence of the coefficient pair (α,η). Determining the value and analyzing their influence are not the point of our study. We conduct the similar analysis of proposition 1 in [5], they show in detail the bound of coefficient α.
>
> [5] Zhang, Yuchen, Percy Liang, and Moses Charikar. "A hitting time analysis of stochastic gradient langevin dynamics." Conference on Learning Theory. PMLR, 2017.
>
> > 7. Line 122: When stating "the teacher predictor satisfies additive expansion," the additive expansion is defined for a probability density, not a classifier.
>
> Sorry for the inaccurate statement. We refer to the confidence predictor g in the teacher model. Its corresponding probability density refers to f in proposition 2.2. It is hard to provide theoretical analysis on the derivative of the neural networks, we can only give the assumption based on the local smoothness of GNN. We could revise the prerequisite of the theorem as "For the GNN in the teacher model, if its corresponding density measure satisfies additive expansion."
>
> > 8. Line 124: Could you explain what is meant by "E_{Y_test}"?
>
> It refers to the expectation taken over the test set. As the reply to your 1st question, the expectation of Err(g) is taken over the test set. Then the inconsistency term A in Theorem 2.3 should be taken from the same range. In the application when the ground truth is unknown, this term could be estimated from the validation set or training set. But the difference between the estimation and the theorem is not the key point in our study.
>
> > 9. Line 145: The first term should change "t" to "t+1".
>
> Yes. According to the definition in Algorithm 1, it should be $CE(g(t)_psi,ŷ_o(t+1))>=CE(g_phi(t),ŷ_o(t))$.
>
> > 10. Line 147: What is the definition of the covariance term in this context?
>
> There are two variables in the covariance term, the cross-entropy and the PL strategy. In this study, we define the PL strategy T as an indicator function of the N PL candidates. The output is 1 for the selected PL samples, and 0 for the non-PL samples. The cross-entropy term also consists of N elements, each of which is the difference between the predicted confidence of the candidates and their ground truth label. The covariance quantifies the relation between the cross-entropy and PL indicator function over these N candidate samples.
>
> > 11. Lines 163-164: There seems to be a typo in the sentence, "we calculate multi-view prediction of the by."
>
> Thanks for pointing out the mistake. It should be  "we calculate the multi-view prediction by the teacher model."
>
> > 12. Do you have any formal proof of Theorem 2.5?
>
> We have shown all the detailed proof of the first inequality in Appendix. It is hard to conduct theoretical proof of the second inequality, as it would be the symbolized representation of the analysis in Line 184-190.

---

> > ### Comment · Reviewer_iJpS · 2023-08-15
> >
> > I have read the authors' rebuttal. I feel they have adequately addressed my questions and concerns. I am increasing my score.

---

### Official Review · Reviewer_nJna · 2023-07-07

**Soundness:** 3 good
**Presentation:** 3 good
**Contribution:** 3 good
**Rating:** 7
**Confidence:** 3

**Summary:**

The article discusses noisy pseudo labeling (PL) on graph data and proposes a new cautious PL methodology (CPL) to improve the graph learning process. The authors conduct experiments to evaluate CPL strategy for link prediction on various datasets and apply it on popular models in node classification task. The result shows that the proposed strategy outperforms other PL strategies. The paper also provides a theoretical analysis of the impact of noisy labels introduced by PL on the graph training procedure. 1. The error introduced by PL is bounded by the confidence of PL threshold and consistency of multi-view prediction. 2. PL can be designed to contribute to the convergence property.

**Strengths:**

The paper makes a highly original and significant contribution to graph learning. It proposes a new cautious Pseudo Labeling (PL) methodology that addresses limitations in prior works by introducing a confidence threshold and a consistency criterion for selecting high-confidence PL samples. This methodology, combined with a new consistency-based PL (CPL) strategy, improves the convergence property of graph learning and outperforms other PL strategies in link prediction and node classification tasks. The research methodology is rigorous, and the paper is well-structured, clear, and provides practical solutions to the challenges of limited and noisy labeled data.

**Weaknesses:**

The paper could be strengthened by addressing several weaknesses. Firstly, in Table 2, the authors can also compare with other PL methods in the link prediction task. Secondly, conducting experiments with larger sample sizes or exploring different configurations would provide a more comprehensive evaluation. Thirdly, clarifying the methodology by providing implementation details and explaining data preprocessing steps would enhance replicability and understanding. Finally, considering the applicability of the proposed approach to different domains with highly imbalanced class distributions or datasets with different types of noise would broaden its practical relevance.

**Questions:**

1. Were there any specific assumptions made when applying the proposed approach to the benchmark datasets? It would be helpful to understand the compatibility of the approach with different dataset characteristics, such as class imbalance or noise types. Insights into these considerations would shed light on the generalizability of the approach.

2. Could you provide additional information on the hyperparameters used in the experiments? Specifically, how were the hyperparameters set for the proposed cautious Pseudo Labeling (PL) methodology and the baseline models? Sharing these details would aid in replicating and fine-tuning the approach in future research.

3. In the discussion of results, could you provide further insights into the potential limitations or failure cases of the proposed approach? Understanding the scenarios where the approach may not perform optimally would help in setting realistic expectations and identifying areas for further improvement.

**Limitations:**

The authors did not mention their limitations.

---

> ### Author Rebuttal · Authors · 2023-08-10
>
> We are most thankful for your thoughtful assessment, and glad to communicate with you on all your concerns:
>
> > Were there any specific assumptions made when applying the proposed approach to the benchmark datasets? It would be helpful to understand the compatibility of the approach with different dataset characteristics, such as class imbalance or noise types. Insights into these considerations would shed light on the generalizability of the approach.
>
> The assumptions of the CPL are the Graph Perturbation Invariant (GPI) property and Additive Expansion Property (AEP). GPI guarantees there is no extreme change during the augmentation process. Some scenarios like the adversarial attack and designed noise are excluded. AEP assumes the continuity of the probability measure in the neighborhood of the local optimal set. The sudden variation in probability density may violate this assumption, such as the highly heterophilous graph. There is no evidence showing that the imbalance and noise type influence the efficiency of CPL. However, their effects on the base model is the key point to the overall performance.
>
> > Could you provide additional information on the hyperparameters used in the experiments? Specifically, how were the hyperparameters set for the proposed cautious Pseudo Labeling (PL) methodology and the baseline models? Sharing these details would aid in replicating and fine-tuning the approach in future research.
>
> CPL is characterized by a relatively small number of hyperparameters. Specifically, there two key hyperparameters that need to be set are the capacity of the number of PL per iteration and the corresponding fine-tuning epochs. These hyperparameters can be chosen based on the available computational resources, as discussed in section 3.2.2 of our analysis. In our experiments, we opted to set the fine-tuning epoch to 20% of the pretraining epoch. The iterative PL capacity results are presented in the following tables.
>
> We have implemented early stop restrictions in our approach. We have defined a lowest acceptable threshold, denoted as Th, and a fine-tuning patience value, denoted as P. In CPL, we employ a top-k strategy. However, if the confidence of a particular sample falls below the threshold Th, it will not be assigned a pseudo-label. The threshold Th can vary depending on the dataset and its specific characteristics. The exact values used for Th are reported in the subsequent section. Furthermore, we have incorporated an early stop mechanism based on the fine-tuning patience P. If the CPL process fails to improve model performance for P consecutive iterations, the training is terminated. In our experiments, we have set P to a value of 10. These early stop restrictions serve to control the quality and effectiveness of PL.
>
> Link prediction:
>
> | Dataset | CiteSeer | Actor | WikiCS | TwitchPT | AmazonPhoto |
> | -- | -- | -- | -- | -- | -- |
> |#PL/itr| 100| 200| 1000 | 300 | 500 |
> | Th | 0.8 | 0.9 | 0.98 | 0.9 | 0.9 |
>
> Node classification:
> | Dataset | Cora | CiteSeer | PubMed | AmazonPhoto | LastFMAsia |
> | -- | -- | -- | -- | -- | -- |
> |#PL/itr| 100| 200| 1000 | 300 | 500 |
> | Th | 0.6 | 0.8 | 0.8 | 0.8 | 0.8 |
>
> > In the discussion of results, could you provide further insights into the potential limitations or failure cases of the proposed approach? Understanding the scenarios where the approach may not perform optimally would help in setting realistic expectations and identifying areas for further improvement.
>
> The proposed model represents a basic implementation of confidence-based PL for multiview graph learning. While it aligns with theoretical analysis, there is ample room for further enhancement. One potential area for improvement lies in the measurement of confidence. In the current CPL approach, we rely on the average of multiview predicted probabilities, which is a biased estimation of confidence. By incorporating a proper confidence estimation that takes into account uncertainty and factors in downstream tasks, we can potentially enhance the underlying model. Another avenue for advancement is the development of more advanced and adaptive PL strategies. For instance, we could consider incorporating a diversity penalty during the selection of PL candidates. This would help to promote a more diverse and representative set of candidates, thereby enhancing the overall learning process.
>
> On a different note, providing an explicit condition for the theorem is challenging. However, it is possible to circumvent the assumptions in specific scenarios. As previously mentioned, the AEP assumes the continuity of probability near the local optimum. Yet, situations characterized by sudden variations in probability density, such as those encountered in heterophily graphs or instances of overconfidence in incorrect samples, may violate this assumption. By acknowledging and addressing these specific conditions, we can refine the model's applicability and effectiveness.

---

> > ### Comment · Reviewer_nJna · 2023-08-18
> > **Post-rebuttal comment**
> >
> > Thank you for the detailed explanations. I am willing to champion this paper.

---

> > > ### Author Response · Authors · 2023-08-19
> > >
> > > Thank you for your positive feedback and for championing our paper. We appreciate the time you dedicated to reviewing our study and engaging in the rebuttal process. Your valuable insights and suggestions have greatly contributed to the improvement of our paper.

---

### Author Rebuttal · Authors · 2023-08-10

The revised main scheme and the required figures of the experiment from  **Reviewer 4kBQ** are shown in the supplementary PDF file.

---

### Comment · Area_Chair_sD2Q · 2023-08-11
**Rebuttals are visible now**

Hi reviewers,

Please take a look at the rebuttals when you have some time. Thanks.

AC

---

### Decision · Program_Chairs · 2023-09-21

**Decision:**

Accept (poster)

**Comment:**

The paper studied pseudo labeling on graph data, namely how to mitigate the negative effect of incorrect pseudo labels. The authors presented theoretical analysis identifying two important factors, pseudo-label confidence and multi-view consistency, and then proposed a practical method based on the theoretical understandings. The paper has strong motivation and contributions, and the four reviewers were all positive. Thus, we should accept the paper for publication.